# Deep transcriptomics reveals cell-specific isoforms of pan-neuronal genes

Zachery Wolfe [1], David Liska [2] & Adam Norris [1] ✉

Profiling alternative splicing in single neurons using RNA-seq is challenging due to low capture efficiency and sensitivity. We therefore know much less about splicing patterns and regulation across neurons than we do about gene expression. Here we leverage unique attributes of *C. elegans* to investigate deep neuron-specific transcriptomes with biological replicates generated by the CeNGEN consortium, enabling high-confidence assessment of splicing across neuron types even for lowly-expressed genes. Global splicing maps reveal several striking observations, including pan-neuronal genes harboring cell-specific splice variants, and abundant differential intron retention across neuron types. We develop an algorithm to identify unique cell-specific expression patterns, which reveals both cell-specific isoforms and potential regulatory factors establishing these isoforms. Genetic interrogation of these factors in vivo identifies three distinct splicing factors employed to control splicing in a single neuron. Finally, we develop a user-friendly platform for spatial transcriptomic visualization of these splicing patterns with single-neuron resolution.

A neuron's development and function are controlled both by the genes it expresses and the isoforms of those genes it selects. Much recent effort has gone into determining the transcriptional states of individual neurons, but analyzing co-transcriptional and post-transcriptional regulatory events such as alternative splicing is a more challenging task[1–5]. The inherent low capture efficiency and sensitivity of single-cell sequencing approaches causes difficulty in accurately calling alternative splicing, especially for lowly- or moderately-expressed genes[6–8]. In addition, the molecular mechanisms by which neurons establish cell-specific isoforms are difficult to identify using current single-cell transcriptomic approaches in isolation.

Although difficult to assess with single-cell resolution, alternative splicing can have profound impacts on neuronal development and function[9,10]. Even relatively small alternatively-spliced exons can have major impacts on neuronal function, for example conferring switch-like gene regulation[11,12] or generating complex functional diversity from a single gene[13–15]. Regulation of alternative splicing is thus critical for the function of individual neurons, but identifying alternative splicing and defining the underlying regulatory mechanisms with single-cell resolution remain major challenges.

We set out to address these challenges using deep transcriptomes from individual neuron types across the *C. elegans* nervous system. Because the developmental trajectory of the *C. elegans* nervous system is invariant across individuals, thousands of the same neurons can be isolated from thousands of isogenic individuals. The resulting deep single-neuron transcriptomes, complete with multiple biological replicates, allow robust analysis of RNA splicing with uniform coverage throughout the gene body and the ability to assay even lowly-expressed genes. Obtaining single-cell splicing patterns in a powerful genetic system like *C. elegans* also allows the next step to be taken: identifying the regulatory mechanisms by which single-cell isoforms are established.

Here we identify neuronal alternative splicing with cell-specific resolution using deep transcriptomes generated by the CeNGEN consortium[16,17]. These global splicing maps reveal several striking observations, including genes that are expressed pan-neuronally but harbor highly cell-specific splice variants, as well as an unexpected

[1]Department of Biochemistry, University of California, Riverside, Riverside, CA, USA. [2]Office of Information Technology, Southern Methodist University, Dallas, TX, USA. ✉e-mail: adamn@ucr.edu

abundance of differential intron retention across neuronal cell types. We develop an algorithm to identify unique cell-specific expression and use it to identify both cell-specific isoforms and potential regulatory RNA binding proteins that establish these isoforms. We genetically interrogate these RNA binding proteins in vivo and identify three distinct regulatory factors employed to establish unique splicing patterns in a single neuron. Finally, we develop a user-friendly platform for spatially visualizing these splicing patterns with cell-specific resolution.

## Results

### Deep transcriptomes reveal cell-specific splicing patterns

To dissect global splicing choices at the single-cell level with high confidence, we took advantage of a valuable resource, the deep cell-specific transcriptomes generated by the CeNGEN consortium[16,17]. The consortium genetically labeled individual *C. elegans* neurons with fluorescent reporters, grew many thousands of isogenic animals, and then dissociated and sorted a population of thousands of labeled neurons (Fig. 1A). These neuron-specific populations were subjected to rRNA depletion, cDNA synthesis, and short-read Illumina sequencing.

The resulting deep-sequenced libraries thus represent relatively large amounts of starting RNA sampled from thousands of genetically identical copies of a given neuronal cell type. Deep sampling permits analysis of genes that are not highly expressed, uniform transcript coverage enables splicing analysis across the entire gene body, and independent biological replicates increase statistical power (Fig. 1A). The consortium has generated deep-sequenced data with multiple biological replicates for 46 of the 118 anatomically-distinct neuron types in *C. elegans*.

Using this dataset, we performed pairwise comparisons of gene expression (using DESeq2[18]) and alternative splicing (using JUM[19]) between all possible pairs of neuronal cell types. In the example in Fig. 1A, we compare alternative cassette exons (exon skipping) between the AVM mechanosensory neuron and the AVL multimodal neuron. Using thresholds of |ΔPSI| (change in Percent Spliced In) > 10% and FDR-corrected *q*-value < 0.05, we identified 75 differential cassette exons. Results for all 2070 pairwise comparisons are shown for cassette exons in Fig. 1A.

Such a systematic global accounting of single-neuron splicing choices enables both gene-centric and cell-centric analysis of alternative splicing. An example of gene-centric analysis is displayed in Fig. 1B, C, focusing on exon 3 of the *unc-31/CADPS* gene, which is an essential mediator of neuropeptide release[20]. The exon encodes 12 conserved amino acids located upstream of the major regulatory and structural domains. Splicing levels vary widely across neuron types, from mostly included (AVL, 78% included) to half included (AVM, 53%) to completely skipped (AVG, 1%). Figure 1C reveals a nearly continuous gradation of splicing levels of this exon across all neuronal cell types. This figure also highlights the power of biological replicates in distinguishing cell-specific measurements that are highly reproducible (e.g., AVL) versus those that are highly variable (e.g., AIY).

Cell-centric analysis of alternative splicing is displayed in Fig. 1A–E and Supplementary Fig. 1A–F. JUM is able to interrogate many modes of differential alternative splicing, as shown in Fig. 1D, including technically challenging categories that are often ignored, such as "composite" events composed of multiple intertwined splicing choices. Comparisons of differential composite splicing across all cell type comparisons are displayed in Fig. 1E, and data across all cell types for each class of alternative splicing are displayed in Supplementary Fig. 1 and Spreadsheet S1.

We asked whether differential cell type expression of various alternative splicing events are correlated with each other. For example, if AVL-AVM comparison reveals many differential alternative cassette exons, are there likewise many differential alternative 5′ splice sites? Indeed, such a comparison across all cell types reveals a strong

correlation (Fig. 1F). Similar comparisons across all splicing types reveal consistently high correlations (Fig. 1G). Intron retention correlations are somewhat lower compared to other splicing types, suggesting the potential for distinct modes of regulation unique to intron retention. Overall, however, differences across cell types are concerted across types of splicing, rather than certain cell types bearing unique splice-type signatures.

In contrast, differences in gene expression across cell types correlate poorly with differential alternative splicing (Fig. 1G, H, Supplementary Fig. 1G, H), indicating that gene expression and alternative splicing patterns are globally distinct and are regulated orthogonally across neuronal cell types. Taken together, this deep neuron-specific transcriptomic data provides a powerful resource for interrogating alternative splicing and gene expression across individual neuron types.

### Widespread differential intron retention across neuronal cell types

In total we identified 15,515 alternative splicing events in 5779 distinct genes (Fig. 2A). Among these alternative splicing events, we noticed a surprising degree of alternative intron retention across neuronal cell types (Fig. 2A–C). Indeed, intron retention accounts for over half (52%) of all differential splicing, followed by alternative 5′ splice sites (27%) and 3′ splice sites (11%). This is surprising, as cell-specific intron retention has historically received less attention relative to other splicing types such as cassette exons. On the other hand, regulated intron retention is a prominent phenomenon in plants[21] and is increasingly appreciated to be important for developmental and stimulus-dependent regulation of neuronal transcripts[11,22–24].

Our data show differential intron retention across neuron types to be very widespread. For example, it is over 7-fold more prevalent than differential cassette exon alternative splicing (Fig. 2B, C). The majority of these intron retention events result in the introduction of frameshifts and/or premature stop codons (Fig. 2D), and the most common scenario is for a given gene to harbor only a single retained intron (Fig. 2E).

We sought to test whether the preponderance of intron retention in our data represents ground truth biological observations or might be inflated by technical artifacts. First, we considered whether intron retention signals might falsely arise from stochasticity in lowly-expressed genes (represented by few sequencing reads) or in technical attributes such as PCR artifacts. However, statistically significant intron retention events tend to be covered by hundreds of reads per biological replicate, and the % included values are reproducible across replicates from a given cell type (Fig. 2D, E). For example, an intron in the *pct-1* gene is consistently spliced out in most neurons (~0% retained) but consistently retained in CAN neuron (69%). This argues against a stochastic origin of the intron retention signal.

Second, we considered whether de novo mutations in the individual worm strains used to sort specific neuron types might result in splice site mutations causing intron retention specific to that strain. This might occur relatively easily in genes that are not essential for viability or growth. However, using the mapped RNA-Seq data to inspect sequences of the retained introns and surrounding exons, we found no evidence for de novo mutations in the intron-retained cell types compared to the intron-spliced cell types (Supplementary Fig. 2A, B).

Third, we considered whether technical attributes specific to the deep transcriptomic experimental procedure (e.g., cell dissociation, rRNA depletion) might result in erroneous intron retention signals not found in samples prepared by other methods. We thus compared our data to RNA-Seq data obtained from wild-type whole animal RNA isolated via polyA selection. An example for the gene *pqn-53* is shown in Fig. 2F, in which the intron is retained at various levels across neuron types, from mostly retained (ASK, 72%) to mostly spliced (RIC, 10%).

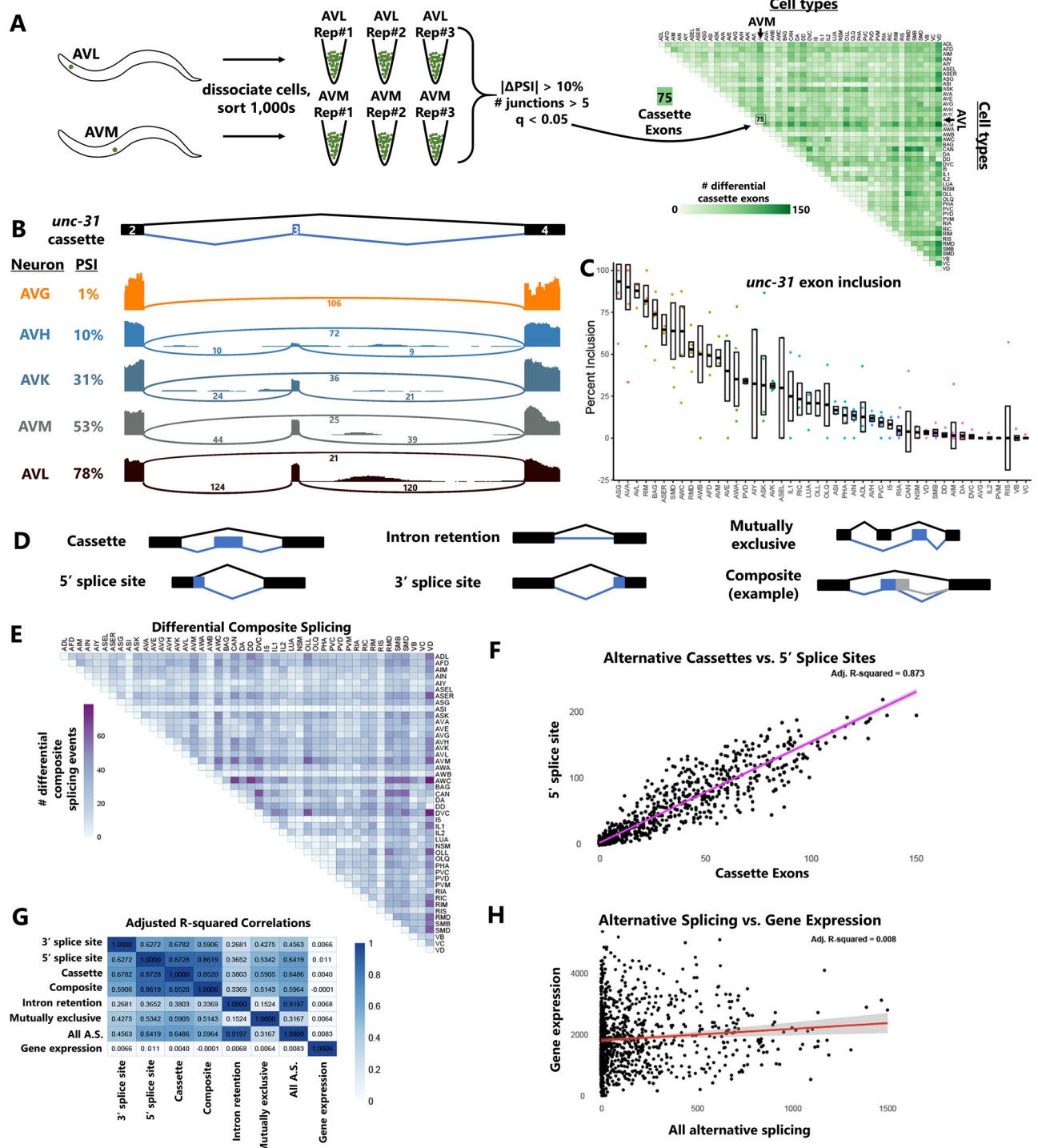

**Fig. 1 | Deep transcriptomes reveal cell-specific alternative splicing patterns.**
**A** Schematic of the analysis pipeline. Deep transcriptomes with biological replicates were obtained by the CeNGEN consortium by sorting large isogenic populations of dissociated worms harboring neuron-type-specific fluorescent transgenes. We then used the RNA-Seq data to call differential splicing across all possible pairwise combinations of neuron types. As an example, AVM vs AVL comparison results in 75 differentially-spliced cassette exons according to our ΔPSI (10%) and *q*-value (0.05) cutoffs. **B** Sashimi plots for a cassette exon (exon #3) in *unc-31*, illustrating gradations of inclusion levels across neuron types. Plots are from single biological replicates; PSI values are means across replicates. **C** Median ± SEM of PSI values

across all neuron types, including each biological replicate value, distinguish highly replicable observations (e.g., AVL) versus variable observations (e.g., AIY).
**D** Illustration of the variety of alternative splicing types accommodated by the JUM analysis pipeline. **E** Widespread differential composite splicing, a type of alternative splicing that is historically difficult to analyze. **F**, **G** Correlation in the magnitude of different gene regulatory events. Each point represents a single-cell vs. cell comparison (e.g., AVM vs AVL). Shaded area = 95% C.I. **H** Summary of correlation values (adjusted R-square) revealing correlation across different types of splicing, but little correlation between gene expression and alternative splicing. Shaded line = 95% C.I.

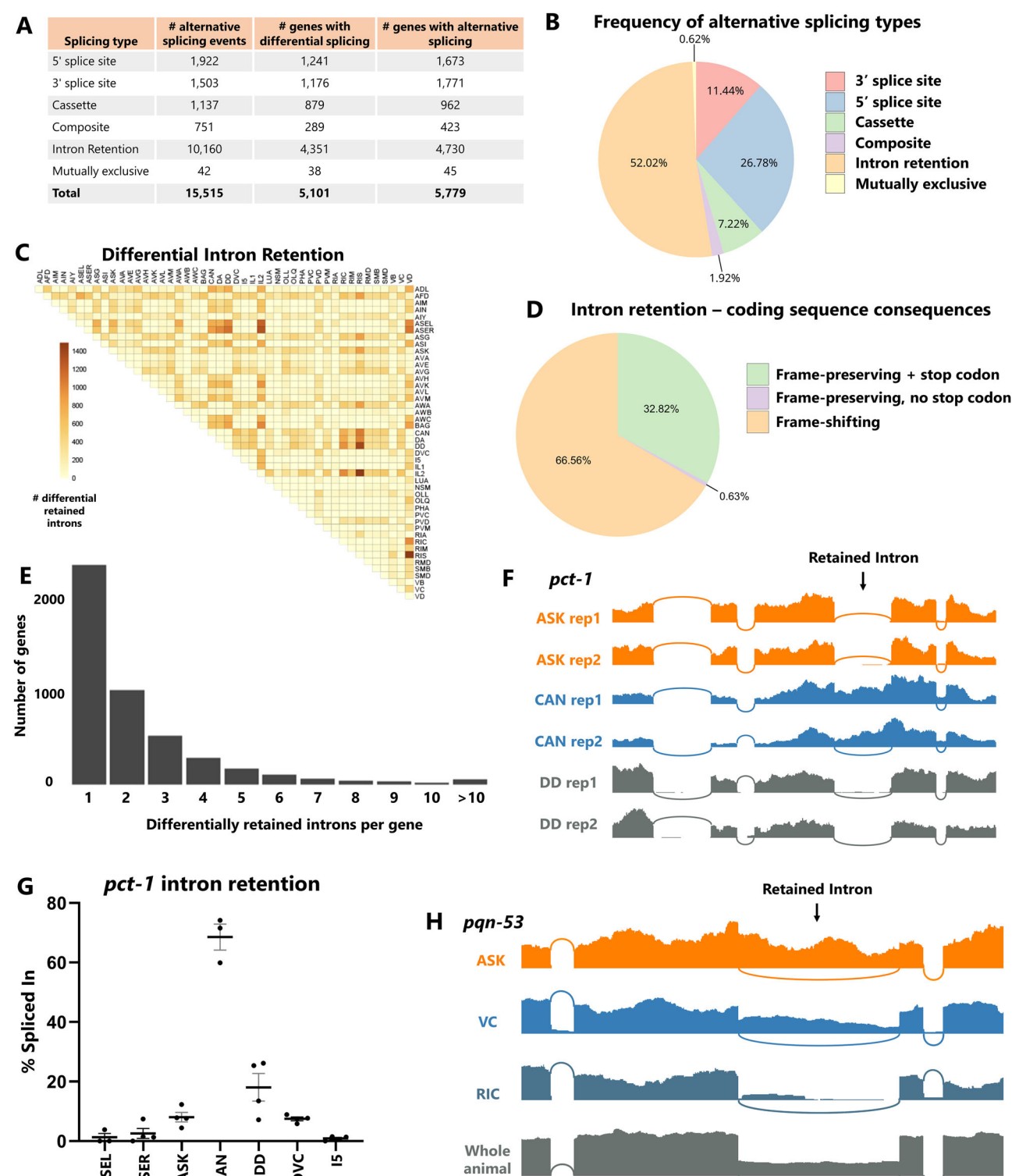

**Fig. 2 | Widespread differential intron retention across cell types. A** Total number of alternative splicing events (defined as PSIs between 10 and 90%) and differential alternative splicing (|ΔPSI| > 10%) detected in our analysis. Minimum of 5 junction-spanning reads per biological replicate. **B** Fraction of differentially-regulated alternative splicing corresponding to different types of alternative splicing. **C** Differential intron retention across individual neuron types is widespread. **D** Fraction of differential intron retention introducing a frame-shift, 1 or more stop codons, or neither of the above. **E** Histogram displaying a number of genes containing various quantities of differentially-retained introns. **F** RNA-Seq coverage showing an example of an intron that is highly retained in CAN but almost completely spliced out in all other neurons. **G** The pattern of *pct-1* intron retention is consistent across multiple biological replicates (individual dots) in various neuron types. *n* = 3, 4, 4, 3, 4, 4, error bars = S.E.M. **H** An example of a retained intron with retention signal in many neuron types, and confirmation that the retained intron is readily apparent from whole animal polyA-selected RNA as well.

Wild-type whole animal RNA-Seq also reveals substantial retention of this intron, thus confirming the observation of alternative splicing of this intron across substantial variations in technical and experimental conditions.

Finally, we sought to validate our intron retention observations with an orthogonal experimental approach. We selected five retained introns with high uniqueness index values and performed RT-PCR on RNA extracted from a population of whole animals. In all five cases, both intron-skipped and intron-retained products are visible (Supplementary Fig. 2C). We therefore conclude that the surprising preponderance of differential intron retention across neuronal cell types is likely a real biological phenomenon. If true, cell-specific intron retention would represent an under-appreciated source of gene regulation at the level of individual neurons, representing around half of differential alternative splicing across neuron types.

## Uniqueness index identifies pan-neuronal genes with unique cell-specific isoforms

A global survey of differential alternative splicing in single neurons allowed us to ask whether genes expressed in many or all neurons might have isoforms restricted to a single neuron or a small number of neuron types. Such isoforms might be unannotated in current databases obtained from data on whole animals but could represent highly specialized isoforms of important neuronal genes with cell-specific functions. To search for such isoforms, we developed an algorithm that aggregates statistically significant differential alternative splicing across all pairwise cell type comparisons. This algorithm, which we refer to as a "uniqueness index," incorporates both statistical significance and effect size to identify splicing events that are robustly different in one cell type compared to all others.

In Fig. 3A, an example is shown for exon #3 of the pan-neuronal gene *unc-31*[25], in which significant splicing differences in AVL versus all 45 other neurons are summed to obtain a uniqueness index. A positive value indicates that the exon is more included in AVL compared to other cells. The maximum possible value of the uniqueness index is 45, which would represent 100% exon inclusion in AVL and 0% inclusion in all other 45 cells. The minimum value is −45, and a value of 0 indicates no systematic difference in splicing compared to all 45 other neurons. Therefore, the uniqueness index of 23 in Fig. 3A indicates a strong exon inclusion signature for the *unc-31* exon in AVL. This is in line with data in Fig. 1B, C showing that the exon is included in AVL and a few other neurons, but skipped in most neurons.

In the case of alternative 3′ splice sites (Fig. 3B), the highest uniqueness value belongs to the gene *mec-2/Stoml3*, which is predicted to select a downstream 3′ splice site in mechanosensory touch neurons (AVM and PVM), while all other neurons select the upstream 3′ splice site (Fig. 3B, row 1). This is precisely in line with our recent work in vivo showing that touch neurons exclusively select the downstream 3′ splice site, which encodes an alternative C terminus, and that this choice is necessary for the function of the MEC-2 protein in touch neurons[15].

The second-highest 3′ splice site uniqueness value (Fig. 3B, row 2) belongs to the gene *dgk-1*, diacylglycerol kinase theta, which is broadly expressed in the nervous system[26]. While most neurons select the upstream 3′ splice site, a small number of neurons also select an unannotated downstream 3′ splice site. The consequence of this alternative splicing choice is two different mRNA 3′ ends (Fig. 3C) encoding either 67 (canonical) or 31 (non-canonical) amino acids. After an initial 6 amino acids of sequence homology, the two alternative isoforms encode distinct amino acid sequences. Interestingly, the cells that uniquely select the downstream splice site (DA and VB neurons) are both excitatory motor neurons, while the neighboring inhibitory motor neurons (VD and DD) exclusively select the upstream splice site (Fig. 3C, Supplementary Fig. 2D).

To test the predicted *dgk-1* splicing patterns in vivo, we generated transgenes in which the upstream splice site is RFP-tagged and the downstream splice site is GFP-tagged (Fig. 3D), driven by the pan-neuronal *rgef-1* promoter. As predicted, most neurons are only RFP+, except for some motor neurons in the ventral nerve cord which are both RFP+ and GFP+. Co-labeling with a BFP expressed in excitatory motor neurons confirms that the neurons expressing both RFP and GFP are excitatory motor neurons, while inhibitory motor neurons only express RFP. Therefore, as predicted by the uniqueness index, a small subset of neurons expresses an unannotated unique isoform of the important broadly-neuronal gene *dgk-1*.

A systematic analysis of uniqueness values for splicing events in pan-neuronal genes reveals widespread cell-specific splicing patterns (Supplementary Fig. 2E), for example in the pan-neuronal kinesin gene *unc-104*[25] which encodes a motor required for synaptic transport via microtubules[27]. The uniqueness index identifies an unannotated microexon that is included only in AVM and PVM neurons, both of which are gentle-touch mechanosensory neurons. The result is that all neurons express UNC-104, but AVM/PVM neurons express an isoform with 7 additional amino acids in the motor domain (Supplementary Fig. 2F). This could represent a useful molecular specialization in touch neurons, which possess unique anatomical features including neurites with unusual bundled microtubule structures[28].

The uniqueness index can be applied to any type of alternative splicing (Supplementary Fig. 3, Supplementary Data 2), even complex composite splicing (Fig. 3E). For example, the neuronal RNA binding protein *unc-75/CELF* gene undergoes composite alternative splicing in which a cassette exon may be included or skipped, and if skipped an alternative 3′ splice site choice is available (Fig. 3F). These splicing choices affect the third RNA Recognition Motif (RRM) domain of UNC-75, encoding either a full-length RRM (exon included), a truncated RRM (exon skipped + upstream 3′ splice site), or no RRM (exon skipped + downstream 3′ splice site). The uniqueness index reveals that in most neurons, the primary splicing choice is cassette exon inclusion, followed by equal amounts of upstream and downstream 3′ splice site choice. However, in two specific cell types (DA and VB motor neurons), exon skipping predominates, and the upstream 3′ splice site is preferentially selected. Thus, the uniqueness index identifies even complex splicing arrangements with signatures unique to individual neuron types.

## Upstream 3′ splice site selection in OLL sensory neurons

While most cells have strong correlations between different types of alternative splicing (Fig. 1G, H), we found at least one strong exception to this pattern in the OLL sensory neuron. The OLL neuron consistently exhibits a large quantity of alternative 3′ splice site selection compared with most other cell types (Fig. 4A, Supplementary Fig. 4A). On the other hand, no such pattern is evident for other splicing types in OLL (see Figs. 1E, F, 2B). Comparing total differential splicing across cell types shows that OLL is a strong outlier for alternative 3′ splice sites, but not for other types of alternative splicing (Fig. 4B).

We asked whether the alternative 3′ splice site usage in OLL follows a consistent pattern (i.e., enriched for upstream vs. downstream splice site usage). Inspecting the uniqueness index values for 3′ splice site usage in OLL compared to all other neurons, there is a strong bias toward positive values (indicating upstream 3′ splice site usage) and very few negative values (Fig. 4C). Indeed, 83% of alternative 3′ splice sites in the OLL neuron show differential upstream splice site usage (Fig. 4D, Supplementary Fig. 4A). The majority (81%) of these alternative splice sites are frame-preserving and free of stop codons (Supplementary Fig. 4C) and are usually a short distance apart (most commonly ≤9 nt, Supplementary Fig. 4D).

An example of an OLL-specific alternative 3′ splice site is shown in Fig. 4E, where a 6-nt alternative 3′ splice site in the ribosomal S6 kinase gene *rskn-1* undergoes primarily upstream splice site selection

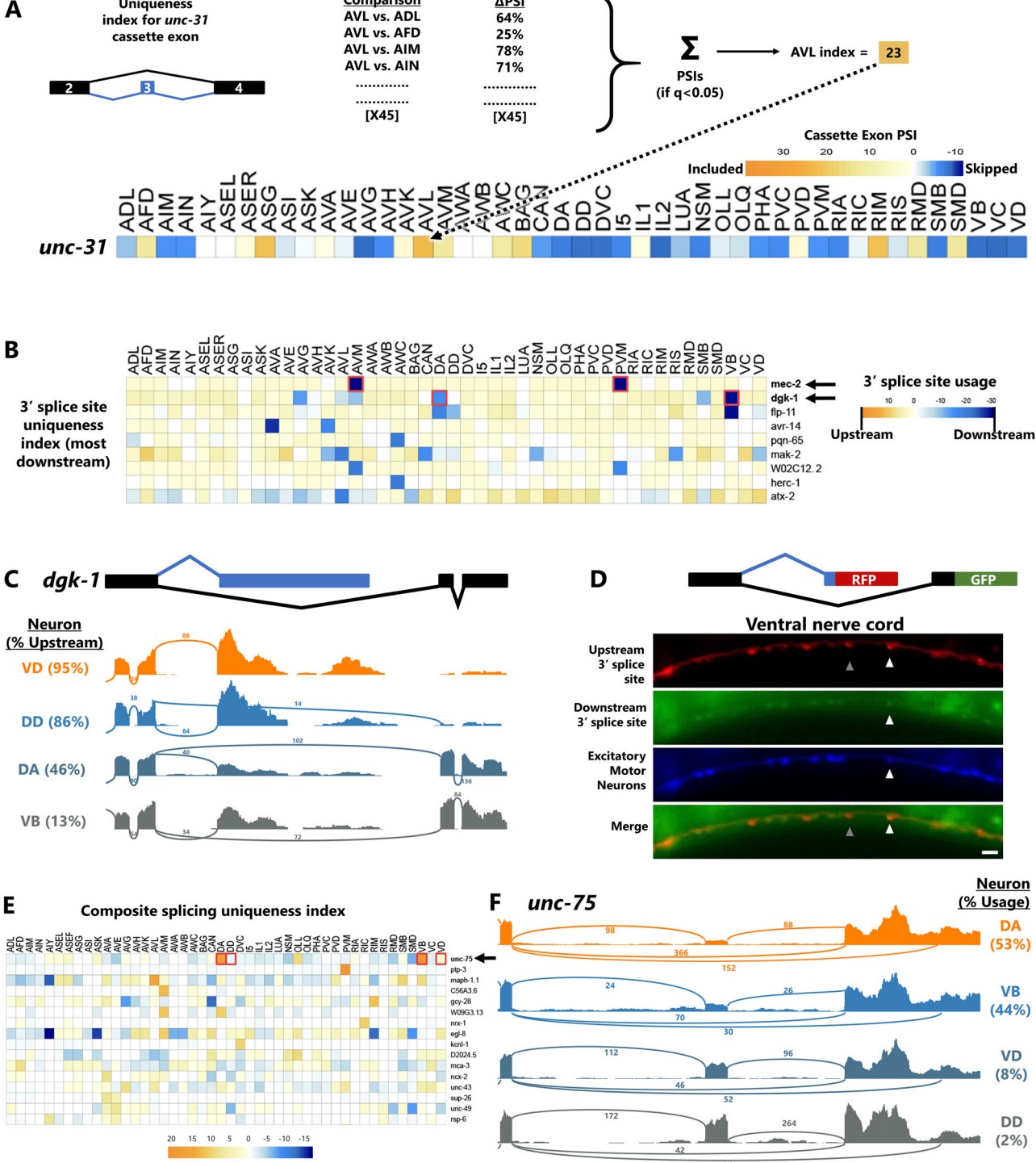

**Fig. 3 | Uniqueness index identifies pan-neuronal genes with cell-specific iso-forms. A** Schematic of the uniqueness index algorithm- for each alternative splicing event, a single neuron (e.g., AVL) is compared against all other 45 neuron types. The sum of all significant ($q < 0.05$) ΔPSIs is the uniqueness index for that cell/splicing-event pair. A score is computed for every splicing event in every neuron type. **B** Uniqueness index for alternative 3′ splice sites, sorted by most "uniquely downstream" splice site choice. **C** Example of *dgk-1* alternative splice site, showing upstream usage in most neurons (e.g., VD and DD) but downstream only in DA and VB. Percentages on left = mean across biological replicates. **D** Transgenic animals harboring pan-neuronal (*rgef-1* promoter) *dgk-1* GFP/RFP splicing reporter as well as cholinergic neuron (*unc-17* promoter) BFP. Scale bar = 10 μm. **E** Uniqueness values are obtainable even for complex splicing patterns such as composite splicing. **F** Read coverage illustrating the top uniquely-regulated composite event, in the RNA Binding Protein *unc-75*. DA/VB neurons skip the cassette exon and select the downstream 3′ splice site, while other neurons (e.g., VD/DD) include the cassette exon and select the upstream 3′ splice site. Plots are from single biological repli-cates; PSI values are means across replicates.

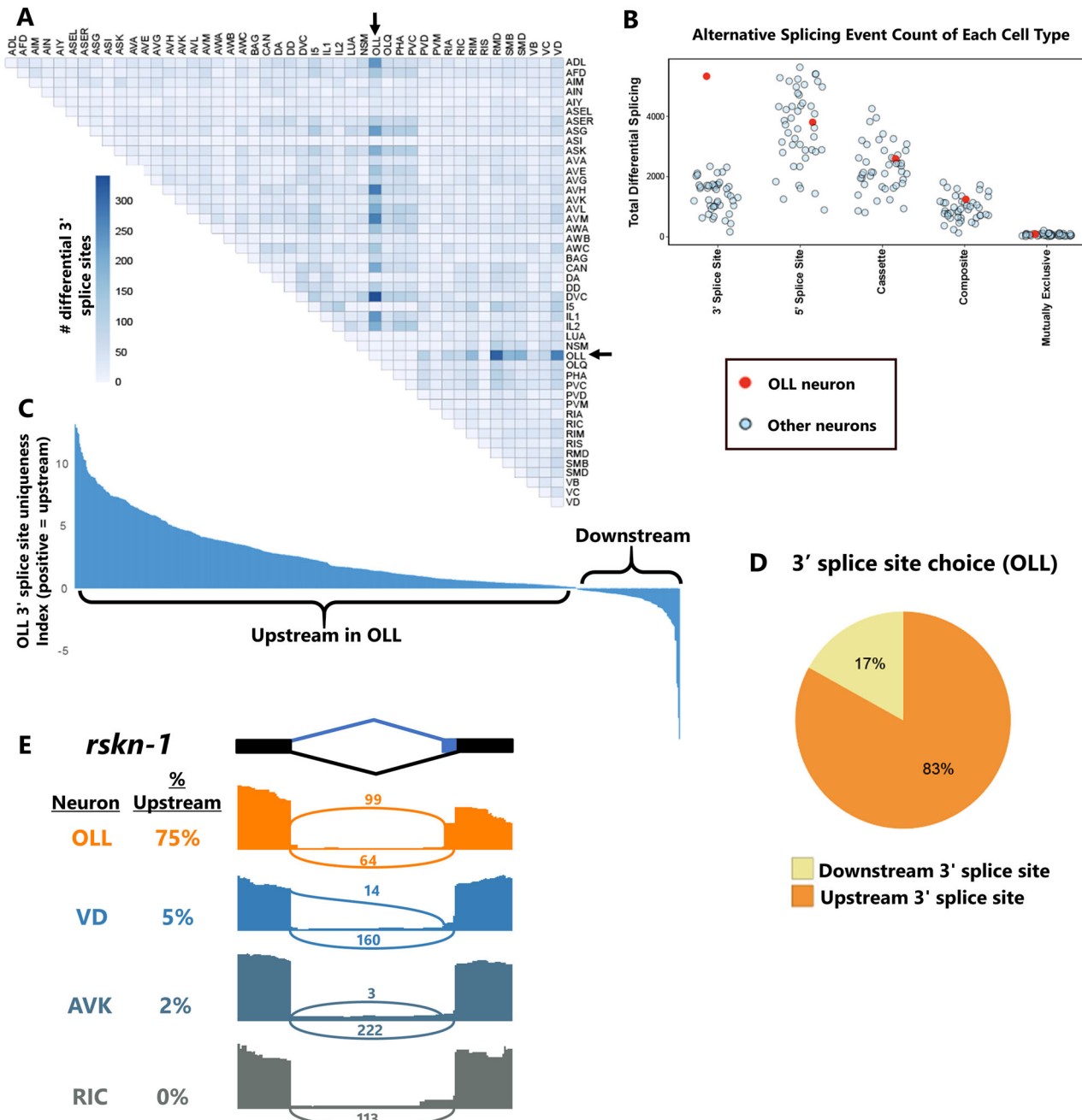

**Fig. 4 | Widespread upstream 3' splice site selection in OLL neurons. A** OLL exhibits abundant alternative 3' splice site choice across cell type comparisons. **B** OLL (red dot) is an outlier in the total number of differential alternative 3' splicing events called across all cell-cell comparisons, but not for other types of alternative splicing. **C** Uniqueness values reveal that OLL is specifically enriched for upstream 3' splice site selection. **D** Summary of statistically significant alternative 3' splice site choices in OLL showing that 83% are biased to upstream selection in OLL. **E** Read coverage illustrating an example of OLL selecting the upstream 3' splice site in the *rskn-1* gene, while other neurons select the downstream splice site. Plots are from single biological replicates; PSI values are means across replicates.

in OLL (75% upstream) but primarily downstream splice site selection in other neurons (0–5% upstream). Together these observations reveal that, at least in the case of OLL, individual neurons possess unique and specific splice-type identities. However, no other cell types or splicing types exhibit such extreme outliers (Fig. 4B) and thus the generality and biological significance of these observations remain unknown.

**Broad applications for uniqueness index and splicing data**

We developed the uniqueness index to identify cell-specific splicing in *C. elegans*, but it is also applicable across various organisms and data types (Fig. 5A, Supplementary Figs. 4 and 5). As one example, we

applied the uniqueness index to gene expression data to identify unique expression signatures in single neurons. Figure 5A shows the top twenty unique genes in the sensory neuron ASK. Strikingly, almost all (80%, 16/20) are "serpentine" G-Protein Coupled Receptor genes, a large family of genes (*srg/srbc/srh/str*/etc.) for which specific ligands are mostly unknown, but many of which encode predicted sensory GPCRs[29]. This is consistent with the role of ASK as a chemosensory neuron[30]. Although none of the top ASK-enriched genes have previously been studied, scRNA-Seq datasets from both embryos and L4 larvae[16,31] corroborate the unique expression of these genes in ASK. In addition to this cell-centric view of unique gene expression, gene-centric analyses can also be performed, searching for which neurons

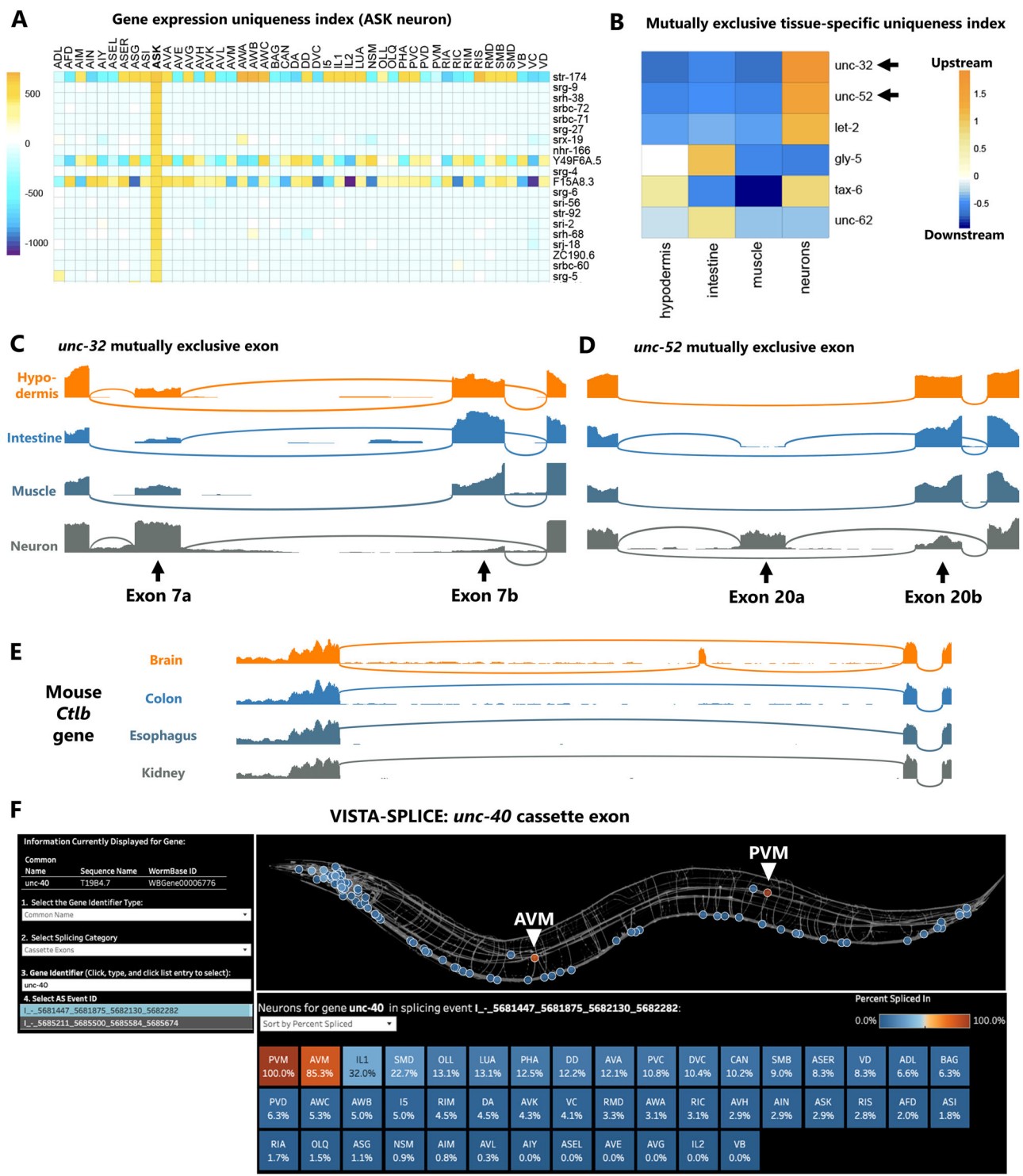

**Fig. 5 | Broad applications for uniqueness index and splicing data. A** Uniqueness index for gene expression values, as determined by DESeq2, sorted here for most unique to ASK neurons. Of the top 20 unique genes to ASK, 80% are serpentine GPCRs (*str/srg/srh/srbc/srx*/etc.) **B** Tissue-specific alternative splicing uniqueness index for mutually exclusive exons, sorted by highest upstream exon selection. The top two mutually exclusive exons are then displayed in (**C**, **D**) as read coverage demonstrating upstream exon selection unique to neurons in both cases. **E** *Cltb* exon 5 is mostly included in mouse brains but is completely skipped in other tissues. **F** Screenshot of the VISTA-SPLICE tool displaying exon inclusion levels for unannotated exon in *unc-40* Netrin receptor gene. Background skeleton of worm modeled after openworm.org. Note that the two touch neurons AVM and PVM display high inclusion levels, while all other neurons display low exon inclusion.

are most enriched for expression of a gene of interest (Supplementary Fig. 4F–K).

As a second example of the utility of the uniqueness index, we applied it to tissue-specific RNA-Seq data[32] to identify alternative splicing choices unique to specific tissues (intestine, muscle,

hypodermis, neurons) (Fig. 5B, Supplementary Fig. 5A–K). Focusing on mutually exclusive exons, we found that the highest-magnitude tissue-specific splicing event is in a gene for the V-ATPase *unc-32* (Fig. 5B). In neurons the upstream exon is selected, while in other tissues the downstream exon is selected (Fig. 5C). This agrees with previous in

vivo reporters demonstrating a selection of the upstream exon specific to the neurons[33]. The second-highest-magnitude splicing event is in the Perlecan *unc-52* gene, which has not been studied in vivo but which displays a similar pattern to *unc-32:* the upstream exon is selected in neurons, while the downstream exon is selected in other tissues (Fig. 5D).

We also sought to apply the uniqueness index to tissue-specific splicing in a different species. Using RNA-Seq data from 13 different mouse tissues[34], we indexed brain-specific genes and isoforms. The top brain-specific genes are all associated with synaptic transmission (*Grm5*, *Gabra5*, *Gabra6*, *Cagnag3*) or other brain-specific functions (*Mobp*, Gpr26, *Lamp5*) (Supplementary Fig. 5L). At the level of alternative splicing, we identified many brain-specific isoforms in which one isoform is expressed in the brain, and the other is expressed in all other tissues (Supplementary Fig. 5M, Spreadsheets S5-6). For example, clathrin light chain B (*Ctlb*) exon 5 is alternatively spliced such that the exon is mostly included in the brain but always skipped in other tissues (Fig. 5E). Together these observations demonstrate the utility of the uniqueness index across different data sources (cell-specific and tissue-specific) and species (worm and mouse).

To increase the accessibility of our neuron-specific splicing data to a broader audience, we developed a visualization platform we call VISTA-SPLICE. This builds on our visualization platform VISTA, in which gene expression values are displayed as a "spatial heatmap" across the *C. elegans* nervous system[35]. Here we adopt the same visualization scheme, except "Percent Spliced In" values are displayed for splicing events of interest. Many splicing types (e.g., cassettes, retained introns, 3′ splice sites) can be visualized. An example is shown (Fig. 5F) for an unannotated cassette exon in the *unc-40/DCC* Netrin receptor. This exon encodes an additional 85 amino acids residing between the P1 and P2 intracellular domains, both of which are important for Netrin-mediated neurite outgrowth[36]. VISTA-SPLICE makes it clear that the exon undergoes cell-specific alternative splicing (Fig. 5F). *unc-40* expression is detected in many neurons, most of which skip the exon (~0% inclusion). But two neurons, the AVM and PVM mechanosensory touch neurons, are clear outliers (85% and 100% inclusion, respectively). Visualizations of such cell-specific splicing patterns should make neuronal splicing analysis accessible to a broader community. The dashboard is freely available at https://public.tableau.com/app/profile/smu.oit.data.insights/viz/VISTA-SPLICEVisualizingtheSpatialTranscriptomeofC_E_/VISTA-SPLICE.

### Identification of causal regulatory factors establishing cell-specific isoforms

A comprehensive cell-specific catalog of alternative splicing in a genetically tractable organism such as *C. elegans* holds promise for going beyond descriptive analysis and proceeding to investigate underlying regulatory mechanisms. To this end, we focused on two of the cassette exons with the strongest uniqueness values but with opposite splicing patterns. First, the *unc-40* cassette exon (see Fig. 5F) which is highly included in touch neurons (AVM/PVM) and skipped in other neurons (Fig. 6A). Second, a cassette exon (currently annotated as a constitutively-spliced exon) in the ligand-gated ion channel *lgc-31* which is ~0% included in AVM/PVM, and ~100% included in other neurons.

We generated transgenic two-color alternative splicing reporters[37] to confirm in vivo the predicted splicing patterns. These transgenic reporters (Fig. 6B) yield RFP fluorescence when an alternative cassette exon is included, and GFP when skipped. As predicted, most neurons are GFP+ (exon skipped) for the *unc-40* transgene, and RFP+ (exon included) for the *lgc-31* transgene (Supplementary Fig. 6A), while in AVM/PVM the patterns are reversed (Fig. 6C, Supplementary Fig. 6B). Expressing the splicing reporters pan-neuronally allowed us to interrogate the splicing patterns in additional neuron types not covered by the CeNGEN RNA-Seq data. For example, we predicted the ALM/PLM neurons (the other two touch neuron cell types) would have the same

unique splicing patterns as AVM/PVM, and indeed this is the case for both *unc-40* and *lgc-31* (Fig. 6C, Supplementary Fig. 6A, B).

The in vivo splicing reporters allowed us to ask additional questions that could not be answered by the sorted neuron data. For example, are the splicing events subject to developmental regulation? Are the cell-specific splicing patterns invariant, or do they exhibit heterogeneity across individual animals? We found that the splicing patterns for both *unc-40* and *lgc-31* cassette exons are invariant across developmental stages and are also invariant across individuals (Supplementary Fig. 6C), indicating that these splicing events are strictly and deterministically regulated at the single-neuron level.

The reporters also allowed us to genetically test in vivo for potential *trans*-acting factors that establish the striking *unc-40* and *lgc-31* splicing patterns. To obtain candidate *trans*-acting factors we returned to our gene expression uniqueness index, this time applying it to differential expression of RNA Binding Protein (RBP) genes. Sorting for uniqueness index values in AVM/PVM should reveal RBPs highly expressed in AVM/PVM compared to all other neurons. The RBPs with the highest uniqueness values were similar when sorted by AVM or PVM (Fig. 6D, Supplementary Fig. 6D), suggesting that these closely related cell types also express similar complements of RBPs. We selected the top five RBPs (Fig. 6D) for further genetic investigation. We previously found two of these RBPs to affect alternative splicing in touch neurons (*mec-8/RBPMS* and *mbl-1/Muscleblind*)[38], while the other three (*sup-12*, *rnp-9*, *D2005.1*) have not previously been reported to be expressed or function in touch neurons.

We systematically crossed both splicing reporters with existing mutants for all five RBP genes[39,40] and tested for changes in cell-specific splicing patterns (Fig. 6E, F). For each splicing pattern, we found a single RBP responsible for establishing the cell-specific splicing pattern. The *unc-40* cassette exon requires SUP-12/RBM24 (Fig. 6E), while the *lgc-31* cassette exon requires MEC-8/RBPMS (Fig. 6F). The remaining three RBPs have no effect on either of the cassette exons. In each case, loss of the regulatory RBP affects all four touch neuron types (ALM/AVM/PLM/PVM) resulting in a splicing switch from the unique touch neuron isoform to the default neuronal isoform and is fully penetrant across individual animals.

Identification of SUP-12/RBM24 as a specific regulator of touch neuron splicing is surprising, as previous work characterized SUP-12 as a muscle-specific splicing factor in *C. elegans*[41,42]. However, our work now identifies additional *sup-12* expression in a small number of neurons (primarily AVM/PVM) and this observation is corroborated by independently generated scRNA-Seq data[16,31]. This indicates that SUP-12 both directs muscle-specific splicing and also directs touch-neuron-specific splicing within the nervous system. SUP-12 thus joins the RBPs MBL-1 and MEC-8 as a trio of factors experimentally demonstrated to provide unique splicing attributes to the touch neurons. Therefore, touch neurons possess at least three non-redundant regulatory mechanisms that each achieve the goal of establishing isoforms in touch neurons that differ from those in the rest of the nervous system.

## Discussion
### Cell-type-specific isoforms of pan-neuronal genes
A useful concept in neuronal gene expression is that of the pan-neuronal versus the cell-type-specific gene. The former constitutes a gene required for fundamental shared processes across all neurons (e.g., synaptic vesicle machinery) while the latter constitutes a gene required for specific properties of a given neuronal type (e.g., neurotransmitter synthesis)[25,43]. Our work highlights additional layers of complexity in this concept, most notably the pan-neuronal gene with cell-type-specific isoforms.

An interesting example of this is in the gene *dgk-1*, which encodes a diacylglycerol kinase that exhibits pan-neuronal expression[26] and is required for the normal function of various neuronal types[44,45]. We identify a novel *dgk-1* isoform that is expressed only in excitatory

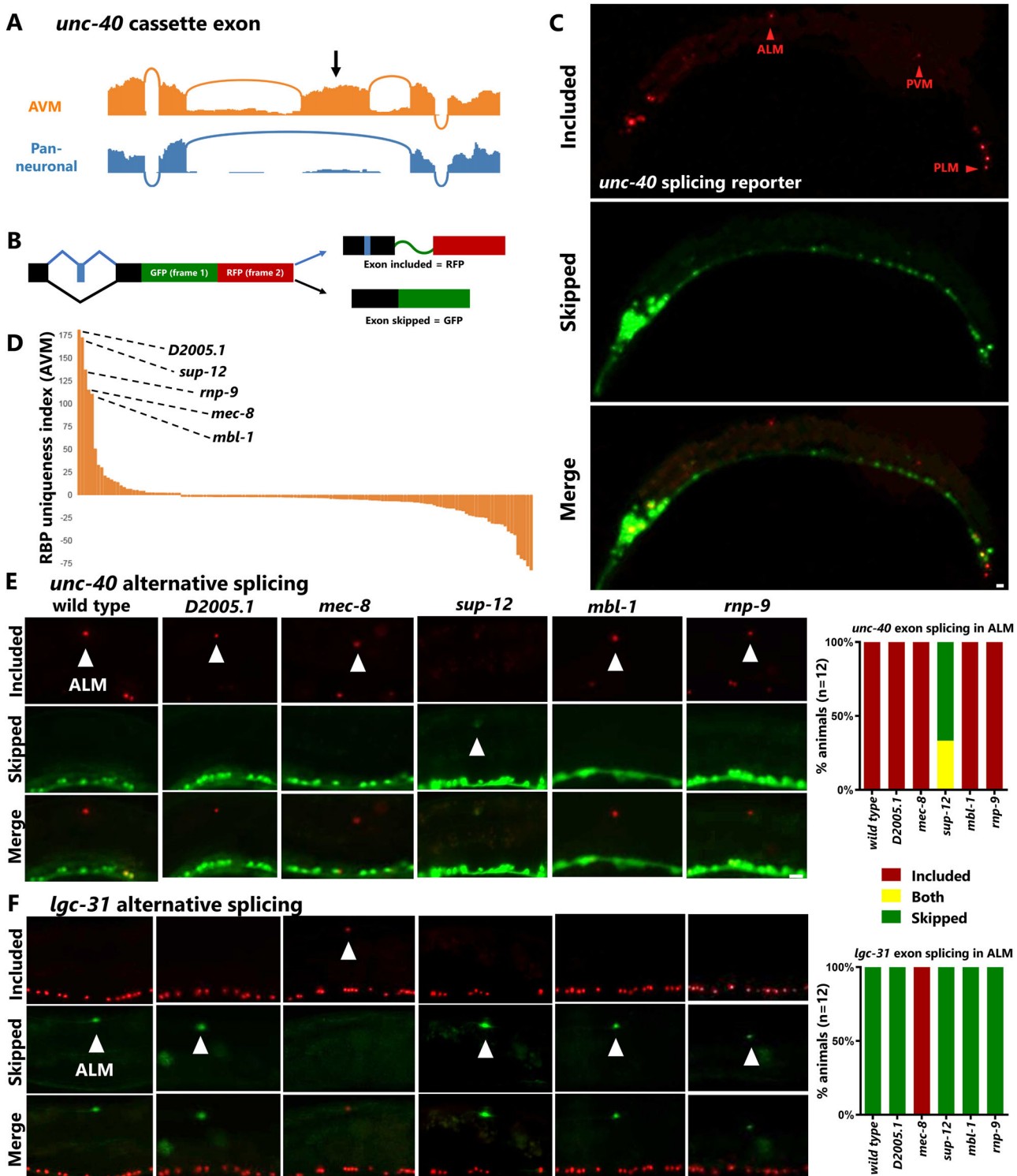

**Fig. 6 | Identification of causal regulatory factors establishing cell-specific isoforms. A** Read coverage for the *unc-40* cassette exon visualized in 5F, showing nearly complete inclusion in AVM neuron, but complete skipping in pan-neuronal RNA-Seq. **B** Schematic for a two-color alternative splicing reporter in which inclusion of a cassette exon is engineered to alter the reading frame, resulting in RFP translation when exon is included and GFP translation when exon is skipped. A Nuclear Localization Sequence directs the localization of both fluorescent proteins to the nucleus to facilitate neuron identification. **C** Splicing reporter for *unc-40* expressed pan-neuronally (*rgef-1* promoter) agrees with the prediction from 5F that AVM and PVM express included isoform, and also reveals inclusion in other touch neurons (ALM/PLM), while most other neurons express only the skipped isoform. **D** Uniqueness index for RBP gene expression in AVM neuron reveals small handful of enriched RBPs. **E** Visualizing exon inclusion in the ALM neuron reveals that SUP-12, but none of the other RBPs, is required for cell-specific *unc-40* inclusion. **F** For the *lgc-31* cassette exon, MEC-8, but none of the other RBPs, is required for cell-specific skipping. All micrographs (**C**–**F**) represent experiments performed 12 times (unique biological replicates). Scale bar = 10 μm.

(cholinergic) motor neurons of the ventral nerve cord, while all other neurons express only the canonical isoform. Therefore *dgk-1* is simultaneously a pan-neuronal gene but is cell-type-specific with respect to individual isoforms. This type of regulatory arrangement might be functionally useful, allowing a single gene to simultaneously perform ubiquitous pan-neuronal function while also providing bespoke features unique to individual neuron types.

## Deep transcriptomes reveal complex and novel neuron-specific alternative splicing

Obtaining deep-sequenced populations of single neurons with multiple biological replicates, coupled with a flexible computational pipeline, allowed us to examine many splicing types that are not commonly assayed at the single-neuron level. For example, we identify widespread composite splicing events regulated across individual neurons. Regulated composite splicing requires a complex coordinated selection of multiple nearby alternative splice sites. Therefore, the observation of widespread composite alternative splicing raises interesting mechanistic questions about how individual neurons simultaneously regulate the selection of multiple splice sites in a single transcript.

We find that intron retention is by far the most common type of differential splicing across neuronal cell types, although intron retention has not historically been a focus of single-cell transcriptomics analyses. The most commonly studied splicing type in single-neuron genomics analyses, including our own, is cassette exons, for both technical and historical reasons[3,4,8,46,47]. Nevertheless, our data here indicate that cassette exons make up only a minor fraction of alternative splicing in the nervous system. With future technical and conceptual innovations, we anticipate an expanding appreciation of the diverse alternative splicing landscape in single cells.

We note that complementary work by the CeNGEN consortium[48] using different analysis pipelines largely agrees with the conclusions presented here (e.g., highlighting the *unc-40* cassette exon in AVM, and lack of relationship between cell-specific gene expression and splicing). On the topic of introns, however, they find less differential splicing (using SUPPA) than we do (using JUM). One likely explanation is algorithm differences: SUPPA depends on transcriptome annotations, while JUM does not. Since the majority of differential retention events we identify are not annotated as alternatively spliced, this likely explains much of the discrepancy. Based on our computational and RT-PCR follow-up analyses, we believe many of these unannotated intron retention events are biologically real and that intron retention is indeed abundant across neurons, thus raising interesting questions about the regulatory and functional implications for individual neurons. Neuronal intron retention is increasingly appreciated as a mechanism for developmental and stimulus-dependent gene regulation, including the phenomenon of "detained" introns which are retained in mature mRNAs until a stimulus causes them to be spliced out, thus resulting in a fully functional spliced mRNA[22–24,49]. Our observations are consistent with such regulatory mechanisms and suggest that intron retention is likely even more widespread than previously anticipated.

## Identifying causal regulatory factors that establish individual neuron splicing choices

A strength of studying single-cell transcriptomes in a strong genetic system such as *C. elegans* is the promise of proceeding from descriptive catalogs of cell-specific isoforms to interrogating the regulatory factors that are responsible. We achieved this goal for two different neuron-subtype-specific splicing events in the touch neurons. RBPs expressed uniquely in these neurons were identified using the uniqueness index, then existing null mutants for the RBP genes in question were crossed with in vivo splicing reporters to test for changes in cell-specific splicing patterns.

One conclusion from these experiments in touch neurons is that there are multiple regulatory routes to the same cell-specific splicing outcome. We identify two different RBPs (MEC-8 and SUP-12) that independently establish unique touch neuron isoforms for different genes. MEC-8 and SUP-12 are potentially direct regulators of *unc-40* and *lgc-31*, as their cognate *cis* elements[50,51] are present in the regions flanking the alternative exon (Supplementary Fig. 6E). Together with our previous work demonstrating a case of combinatorial regulation by MEC-8 and MBL-1 resulting in cell-specific splicing of the *sad-1* kinase[38], at least three different unique strategies are deployed to achieve the same outcome (i.e., touch-neuron-specific isoforms of broadly-expressed genes). As such, there does not seem to be a single conserved master regulator of touch neuron isoforms. Rather, various factors are deployed that are each readily amenable to the task due to their unique expression in this cell type. It will be interesting to test whether such distributed regulatory networks are common across other neuronal cell types as well.

We anticipate that the experimental framework established here will broadly apply to questions of cell-specific gene regulation. For example, the uniqueness index pipeline we used to identify splicing regulators in touch neurons should also be applicable to identifying transcriptional regulators in specific neurons or tissues, or for nominating candidate chemosensory GPCRs in specific neurons (Fig. 5A). We also hope that visualizing the spatial transcriptome with VISTA-SPLICE will lower the barrier to analyzing cell-specific splice isoforms, and the global alternative splicing values reported here will enable researchers to study cell-specific splicing of their genes of interest with ease.

## Methods

### Strains used

*adnEx10[rgef-1::dgk-1 splicing reporter + unc-17::BFP], adnEx11[rgef-1::unc-40 splicing reporter], adnEx12[rgef-1::lgc-31 splicing reporter], D2005.1(ok2689)* I, *mbl-1(csb31)* X, *sup-12(ok1843)* X, *rnp-9(ok144)* X, *mec-8(csb22)* I. Some strains were provided by the CGC, which is funded by the NIH Office of Research Infrastructure Programs (P40 OD010440).

### Imaging

Fluorescent Images were taken with 10× or 20× objectives using a Zeiss Axio Imager.Z1 AX10 microscope and Zeiss ZEN2.5 (blue edition) software. Images were processed using ImageJ 1.54d (NIH, USA. http://imagej.org).

### Splicing reporters

Two-color splicing reporter backbones were obtained as described previously[46,52]. For the inserts, *lgc-31* a 431 bp fragment containing the alternative exon and flanking introns/exons was generated (WBcel235 coordinates chrV:15,513,325-15,513,755), and likewise for *unc-40*, which was an 890 bp fragment (coordinates chrI:5,681,422-5,682,311). For the *dgk-1* 3′ splice site selection two-color reporters, two different splicing reporters were generated. In each case, the fluorescent protein was fused immediately upstream of the respective stop codon.

### RT-PCR

RNA was extracted from mixed-stage wild-type animals via Trizol and Zymo RNA Quick-RNA Miniprep kit according to manufacturer instructions (Zymo R1054), followed by RT-PCR using NEB Lunascript one-step RT-PCR kits (NEB E1555L). In addition to 10+ minutes of DNAse treatment, when possible primers were designed to overlap exon-exon junctions or reside across a large intron to ensure that genomic DNA was not amplified (this was possible in 3 of the 5 cases: *mec-8*, *pqn-53*, and *K03B4.4*).

## VISTA-SPLICE

As with VISTA, upon which VISTA-SPLICE was built, VISTA-SPLICE was created with Tableau and is freely available via Tableau Public. Original anatomical and cell-position information was derived from Open-Worm (openworm.org) and WormAtlas (wormatlas.org).

## Cell and tissue types

SRA run selector[53] was used to search for and download SRR files associated with PRJNA952691[17]. A total of 180 replicates from 46 unique cell types were used; each cell type represents a different neuron in *C. elegans*. SRA run selector was also used to search for and download SRR files associated with PRJNA400796[32]. A total of 22 replicates from 4 unique tissue types were used. The CeNGEN RNA-Seq pipeline from which this data was obtained involved dissociation of whole L4 animals, sorting of GFP (and/or RFP) positive populations of cells, RNA extraction, followed by rRNA depletion, cDNA synthesis, and short-read Illumina sequencing.

## DESeq2/HTSeq

Gene-specific counts were tabulated for each sample using HTSeq[54] and statistically significant differentially expressed transcripts were identified with DESeq2[18]. Identification of uniquely dysregulated genes was determined by filtration of genes using strict and conservative thresholds. For gene expression, including the expression of RNA binding proteins, this threshold was $p$-value $< 0.01$ and $|\log_2 FC| > 2$ as determined by DESeq2. All scripts used to filter and quantify DESeq results can be viewed at https://github.com/xcwolfe/Differential-Expression-in-C-elegans[55].

## STAR

Libraries were sequenced and aggregated by Barrett et al. (PRJNA952691) and Kaletsky et al. (PRJNA400796), then aligned to the worm genome (version WBcel235/ce11) using STAR[56]. STAR parameters include a read overhang minimum of 15 bases and a multimap maximum of 20 locations.

## JUM

The Junction Usage Model (2.0.2) (JUM) was used to identify differentially-spliced isoforms in different cell types and quantify their expression levels by computing the ΔPSI (difference of Percent Spliced Isoform)[11]. JUM reports six different categories of alternative splicing events: cassette exons, mutually exclusive exons (MXE), alternative 5′ splice sites (A5S), alternative 3′ splice sites (A3S), intron retention events, and composite events (a combination of two or more of the previous five categories).

The parameters for JUM A include: --JuncThreshold 5 --Condition1_fileNum_threshold ($n$-1) --Condition2_fileNum_threshold ($n$-1) --IRthreshold 5 --Readlength 100 --Thread 3 (in which $n$ = the number of replicates for a given cell type). The parameters for JUM B include: --Test $p$-value --Cutoff 1 --TotalFileNum 8 --Condition1_fileNum_threshold ($n$-1) --Condition2_fileNum_threshold ($n$-1) (in which $n$ = the number of replicates for a given cell type). The parameters for JUM C include: --Test $p$-value --Cutoff 1 --TotalCondition1FileNum ($n$-1) --TotalCondition2FileNum ($n$-1) --REF refFlat.txt (in which $n$ = the number of replicates for a given cell type). All scripts used to filter and quantify JUM results can be viewed at https://github.com/xcwolfe/Differential-Expression-in-C-elegans.

## Principal component analysis

To create a Principal Component Analysis (PCA) plot of differential gene expression among all 46 cell types, we categorized all cell types by their neuronal function (interneuron, motor, sensory, polymodal). DESeq results were variance-transformed using the vst() function (DESeq2 package)[18] and a PCA was then created using the prcomp() function (stats package) with the center = TRUE argument. The PCA

was plotted using the ggplot2 package[57]. For each category alternative splicing (excepting composite, due to its complex nature) the % inclusion (or % upstreamness) values of all cell types within the PSI matrices were standardized using the scale() function (base package). A PCA was then created using the prcomp() function (stats package) with the center = TRUE argument. The variance of each principal component was calculated by squaring the standard deviation. PCAs were plotted using the ggplot2 package[57]. An additional PCA was created to represent 5 splicing types combined – A3S, A5S, cassette, intron retention, and mutually exclusive exon events (composite splicing events were excluded from this PCA due to over-representation) using the same methodology.

## Summation/quantitation of JUM results

Identification of unique alternative splicing events was determined by filtering events using strict and conservative thresholds. This includes a junction threshold of 5, $p$-value $< 0.05$, $q$-value $< 0.05$, and $|\Delta PSI|$ $> 10\%$ as determined by JUM. For our downstream analysis of alternative 3′ splice site (A3S) and alternative 5′ splice site (A5S) events, we excluded events that contained more than two potential 3′ and 5′ coordinates, respectively.

## Global quantification of differential alternative splicing types

The parameters used to determine the total counts of unique alternative splicing events included (in addition to the significance thresholds listed above) a minimum threshold of 5 counts of alternative splicing events as detected by JUM. This threshold is also applied to the total counts of uniquely spliced genes. Individual cell type vs. cell type comparisons and tissue type vs. tissue type comparisons (i.e., JUM heatmaps Supplementary Fig. 1A−F) retained the minimum threshold of a single (1) count since it is not possible for the same alternative splicing event to be detected more than once in a comparison.

## Uniqueness index

In order to obtain a useful summary statistic from the typical pairwise comparison made by JUM (i.e., 2 different cell types) that would allow the global analysis of splicing events across all available cell types, JUM was run 2082 times – once for each cell type compared to all other cell types ($46 \times 45 = 2070$) and once for each tissue type compared to all other tissue types ($4 \times 3 = 12$). The results of all JUM cell type-cell type comparisons for each alternative splicing (AS) type were merged into a set of large data frames composed of every single AS event detected in any of the 2070 runs. Additionally, a second set of data frames was composed for every single AS event detected in any of the 12 tissue type vs. tissue type runs. For alternative 5′ and 3′ splice sites, as well as mutually exclusive splicing, the most upstream splicing choice was defined as a positive PSI value and downstream defined as negative. When an alternative splicing event was detected in more than one comparison, the ΔPSI value of every identical AS event in all comparisons was added to one another – this number was referred to as the uniqueness index, aka the "sum of all statistically significant ΔPSIs" for a given cell/tissue type and given AS event ID. To ensure that ΔPSI represented the most upstream coordinate/exon choice for all AS events, the ΔPSI of A3S events on the positive (+) strand as well as the ΔPSI of A5S events on the negative (−) strand were reversed during the summation of results.

For the pan-neuronal uniqueness index, we first needed to define which genes are considered pan-neuronal between the cell types in our analysis. An initial list of genes was extracted from the gold-standard list of pan-neuronal genes, as defined by Stefanakis et al.[25]. From this list, out of 126 possible neuron types, the median number of neurons in which these gold-standard genes were expressed was 123 with a mean of 114.39; therefore, the threshold for pan-neuronal expression in our analysis included genes that were expressed in at least 123 out of 126

neuron types according to CeNGEN single-cell data[16]. A total of 352 genes met this median cutoff threshold. We then proceeded to calculate the absolute value of the sum of the first ΔPSI values for all cell types of each significant alternative splicing event within each of these genes. This approach captures both the highest and lowest sum of the first ΔPSI values. Finally, we ordered the genes from highest to lowest absolute value (in any cell type) to create the pan-neuronal uniqueness index.

### Usage of alternative splicing junctions by individual cell types

To determine the splicing junctions used by individual cell types in a non-comparative fashion, we analyzed the detailed output files provided by JUM. Each detailed output file provides the percentage usage of each AS junction for each individual replicate – percentage usage values are consistent between JUM comparisons regardless of the cell types being compared. The percentage usage values of each AS event ID were averaged between replicates of the same cell type and aggregated into their own data frames. These percentage usage data frames were merged together by common AS event ID to create a matrix for each AS type (6 matrices in total, Supplementary Data 3 and 4). Junctions representing percent inclusion levels (for cassettes or introns) or upstream splicing levels (for 5′, 3′, and mutually exclusive splicing) were selected for each splicing event ID.

### Reporting summary

Further information on research design is available in the Nature Portfolio Reporting Summary linked to this article.

### Data availability

The data supporting the findings of this study are available from the corresponding authors upon request. All sequencing data used were previously published, and available at the following GEO accession numbers: PRJNA952691, PRJNA400796, PRJEB22693. Source data for the figures and Supplementary Figs. are provided as a Source Data file. Source data are provided with this paper.

### Code availability

All code produced is available at https://github.com/xcwolfe/Differential-Expression-in-C-elegans.

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

## Acknowledgements

Funding was provided by the National Institute of General Medical Sciences of the National Institutes of Health [R35GM133461] to A.N.; the National Institute of Neurological Disorders and Stroke of the National Institutes of Health [R01NS111055] to A.N.

## Author contributions

Conceptualization: A.N. Data curation: Z.W., D.L., and A.N. Formal analysis: Z.W., D.L., and A.N. Funding Acquisition: A.N. Investigation: Z.W. and A.N. Visualization: Z.W., D.L., and A.N. Writing – original draft: Z.W. and A.N. Writing – review and editing: Z.W., D.L., and A.N.

## Competing interests

The authors declare no competing interests.
