## [Transparent Peer Review file · Nature Communications]

Deep Transcriptomics Reveals Cell-Specific Isoforms of Pan-Neuronal Genes

Corresponding Author: Dr Adam Norris

Version 0:

Reviewer comments:

Reviewer #1

(Remarks to the Author)

This manuscript by Wolfe et al. presents the identification and characterization of neuron-specific splicing events in the *C. elegans* nervous system. Using RNA-seq data for individual neuron classes generated by the CeNGEN consortium, the authors profile alternative splicing patterns across 46 neuron classes and provide a number of interesting observations along with a framework to identify RNA-binding proteins that regulate some of these splicing events. To visualize splicing patterns with single-neuron resolution, the authors developed the web application VISTA-SPLICE, where percent spliced-in (PSI) values can be viewed for alternative splicing events of interest across the different neuron classes.

First, the authors perform pairwise comparisons of gene expression and alternative splicing between each possible neuron-type pair to identify over 15,000 alternative splicing events in ~5,000 genes that are differentially spliced across neurons. The authors explore the data in both a “cell-centric” and a “gene-centric” manner and show a few selected examples of different types of alternative splicing events. E.g. “gene-centric” patterns are exemplified by showing that inclusion/exclusion rates of exon 3 of *unc-30* vary widely across neuron types. The authors highlight an extreme example of “cell-centric” analysis by showing that OLL sensory neurons have increased amounts of upstream 3’ss selection compared to all other neuron types in their analysis.

The most surprising observation from the classification of different types of splicing events is the authors’ claim that intron retention accounts for 52% of all differential splicing in the neuron types they profile.

To further explore the data, the authors propose a “uniqueness index” that aims to measure how broad or neuron-specific is a given alternative splicing event. Using this measurement, they identify some highly specific events. For example, a DA and VB neuron-specific isoform of *dgk-1* that uses an unannotated 3’ss. This uniqueness index is also applied to gene expression data from single neuron classes, but also tissue-specific RNA-seq data from *C. elegans* and mice, to identify top unique genes and splicing patterns in each context. An interesting observation from this analysis is that panneuronal genes can have neuron-specific splicing patterns.

Finally, the authors show in a proof of principle analysis that the combined gene expression and alternative splicing data can be used to identify a trans-acting protein factor that contributes to that specific gene-specific splicing event. For this they make use of *in vivo*, fluorescent splicing reporters and RNAi of candidate RNA binding proteins to begin to generate biological models for how neurons achieve such complex splicing programs.

Overall, the work presented here provides a useful resource to more deeply study neuronal transcriptome regulation by alternative splicing. Having a website to explore the data is also great. The authors are careful to not overstate hypotheses generated by their data and acknowledge technical differences in data analysis compared to Weinreb et al. (Co-submitted). We have a number of points that we think are important to make this resource more broadly useful.

Main points:

1. Even though the authors did not generate the data, it is important that the libraries be described at the beginning, to understand what type of RNA sequencing data this was. Specifically, that this is full length, ribosome-depleted, short read sequencing, etc.

2. One of the main points in our opinion is that, primarily for the intron retention cases, but also more generally, the authors don't really discuss what is the consequence of these events for protein production. In the case of intron retention, would this most often lead to truncated proteins? This is important information to even begin to interpret the relevance of these events. A more general analysis of how these events impact protein output is essential.
3. The authors do not validate any of the intron retention events *in vivo*. This is somewhat problematic given the high number of intron retention events the authors report, and how this is different from the number identified by Weinreb et al. The authors also comment on how their numbers are different and attribute this to the use of different algorithms. However, from both papers, it is unclear whether the authors think one approach has more false positives or false negatives. Perhaps the authors can intersect both sets of called events and collectively validate some of these events.
4. One of the main interesting points the authors highlight is the identification of pan-neuronal genes that have cell-specific expression patterns. However, in its current form this analysis is anecdotal, with only a couple of examples being presented. It seems like the authors could do a more global analysis with the available data: e.g. take genes expressed pan-neuronally and display their splicing uniqueness index in a heat map; or plot the correlation between expression uniqueness and splicing uniqueness; or some other form of exploring this more systematically.
5. Related to the specific examples that they showcase for panneuronal genes that are cell-specifically alternatively spliced, *unc-31* is indeed panneuronal (the authors could cite Stefanakis et al 2015 which is probably the most thorough examination of this expression pattern). For *dgk-1*, the evidence is less obvious from the cited study, and in fact, in the single cell cengen data this does not look panneuronal. The authors could examine this more directly or choose another better example. The validation for the cell specific *dgk-1* splicing is somewhat misleading as the authors drive the reporter with a panneuronal promoter (*rgef-1*) as opposed to using the endogenous promoter.
6. The prevalence of upstream 3' splice site selection in OLL neurons would also benefit from some additional analyses. Specifically, it would be important to show a distribution of % upstream usage to understand whether *rskn-1* is an outlier or whether other mRNAs are also preferentially spliced with high percentages. And second, the consequence for the proteins is not sufficiently explored. In the *rskn-1* case addition of two amino acids may or may not be consequential, and how about for other events? Do they mostly maintain frame and add a small number of aa? or do they cause early stops?
7. The authors highlight the application of the "uniqueness index" they calculate but there is one application that does not seem as developed as it could be, and would be useful if it was: instead of generating a neuron centric uniqueness index as they do for ASK in Fig. 5A, could the authors generate a gene centric, ranked table, that would also make it easy to see in which neurons a particularly specific gene is expressed in? This would be a useful resource to find neuron specific/restricted genes more easily. In addition, given that so many reporters already exist and have been analyzed with single neuron resolution, this would serve as validation for the utility of their index.

Minor points:

1. Microscopy images should be zoomed in to be able to see cell bodies and projections more clearly, they are all a bit blurry.
2. For the *pqn-53* retained intron event, perhaps the authors could verify *in vivo* with an appropriate fluorescent reporter.
3. In paragraph 3 of the "Upstream 3' splice site selection in OLL sensory neurons", there is a typo in "ribosomal"
4. Perhaps the authors could comment on the sequences preceding the upstream 3' splice sites they identify? Do they follow the consensus "UUUCAG" motif seen for *C. elegans* 3'ss?
5. For the cassette exons identified with the strongest uniqueness values (*unc-40* and *Igc-31*), are there consensus RNA-binding motifs associated with RBPs identified to regulate the splicing of these exons (*SUP-12* for the *unc-40* cassette exon and *MEC-8* for the *Igc-31* cassette exon)? If so, are these motifs present in the respective transcripts?
6. In the top panel of Fig. 6C, where is the AVM neuron? In general, the micrographs would benefit from outlining the worm and the cells of interest, and providing close-up views.
7. In the last sentence of the Figure 1 description, there is a typo "between gene expression and alternative splicing"

(Remarks on code availability)

Reviewer #2

(Remarks to the Author)

(Remarks on code availability)

Reviewer #3

(Remarks to the Author)

Wolfe et al report transcriptome analysis of *C. elegans* neurons at single cell and splicing isoform resolution. The authors take advantage of deep cell-specific transcriptomes from the CeNGEN consortium to assess gene expression and alternative splicing features of 46 or the 118 anatomically-distinct neuron types in *C. elegans*. They report clear differences in splicing isoform ratios in different neuron types, concerted across splicing types and distinct from differences in gene expression. Strikingly, intron retention emerges as the most populated category of differentially spliced types. Next they develop a uniqueness index approach that allows to identify pan-neuronal genes expressing unique isoforms in specific

neurons or subsets of neurons, providing interesting biologically interesting examples of this. Intriguingly, they observe that OLL sensory neurons display a very specific pattern of 3' splice site choice compared to all other neuron types whereby more alternative 3' splice site selection events occur and of these, the upstream 3' splice sites are selected in a variety of genes. The authors also illustrate the utility of their uniqueness index framework for assessing gene expression and splicing in other organs and organisms, and specially in the spatial tool VISTA-SPLICE which shows the distribution of splicing choices in neurons along with their localization in the worm body. Finally, using an elegant combination of bichromatic reporters and mutant lines, they identify two RNA binding proteins that regulate inclusion of two alternative exons specifically in touch neurons.

Assessing alternative splicing at single cell resolution has been fraught with controversy due to artifacts in amplification of tiny amounts of RNA that can prevent accurate quantification of isoforms. The authors have taken advantage of the highly uniform development of *C. elegans* to isolate RNA from identical neurons from many animals, thus overcoming the issue of RNA amounts limitation. The paper, along the co-submission by Weinreb et al, provides a highly valuable resource to identify cell-specific splicing variants, compare their patterns of expression and visualize the localization of the neurons that express them. The authors illustrate the value of their approaches, including the identification of isoforms specific of particular neuron types and the discovery that an RNA binding protein previously thought to regulate muscle-specific splicing also serves to regulate splicing of in touch neurons.

In my opinion the following revisions would help to improve the manuscript:

1. While the authors' approach takes advantage of the unique uniform developmental plan of *C. elegans*, it is not possible to discern, in the cases in which exons/introns/splice sites are not fully used or fully skipped, that individual cells do not have a distribution of PSI values because at the end the analysis is carried out in a population of cells (arguably identical, but that is exactly the point, they may not be so). It would be would to discuss this limitation in the text.
2. Figure 1A (and other similar analyses): the results are very interesting but no clear structure emerges from the pairwise comparisons. It may be useful to run PCA analyses to see whether there are clusters of neurons that differ more than others in differential splicing patterns (general or type-specific) and whether such clusters bear any relationship with other developmental, functional, anatomical, etc. criteria.
3. Figure 1G: the message emerging from these analyses is quite intriguing, but intron retention (the largest category of differential events according to the results of Figure 2) seems to be much less "coordinated" with other splicing types, particularly with mutually exclusive alternative exons. What do the authors make out of this?
4. The results of Figure 2 are quite striking, but it would be good to validate the extent of intron retention in a few genes / neuron types by an independent method (e.g RT-PCR) because this is the event type that typically shows lower validation rates. I would suggest to discuss the concept of detained introns (PMID: 25561496), which correspond to single introns that are developmentally regulated, remain associated to the transcription site and are removed upon reception of a signal to allow rapid induction of gene activity. This might be the case for at least some of the intron retention events identified by the authors. In this regard, how often more than one intron retention events are observed within the same gene? Figure 2C: do the numbers in the key of the heatmap correspond to the number of intron retention events?
5. The authors convincingly argue about the value of uniqueness indexes, but it would be interesting to know more about the general distribution of uniqueness index values across neuron types and alternative splicing events. The authors are in a unique position to dive into these data. For example, what fraction of neuron types harbor highly distinct uniqueness indexes for particular classes of alternative splicing events. Is the case of OLL neurons and alternative 3' splice sites quite unique? As in point #2 above, is there a structure emerging from the data?
6. The results of Figure 6 nicely illustrate how the dataset can be used to infer mechanisms of regulation. Are other examples of alternative splicing events likely to be regulated by *mec-8* or *sup-12* in touch neurons? i.e. events differentially regulated in this neuron type with predicted binding sites for these factors (e.g. PMID: 26430555, PMID: 28003515).

(Remarks on code availability)

Version 1:

Reviewer comments:

Reviewer #1

(Remarks to the Author)

The authors have done a thoughtful job addressing the points we raised. We think the manuscript is improved and this will be a very useful resource for many researchers.

A minor suggestion that we think would improve readability is to label the panels in the supplementary Figures A, B, C, etc. At the moment the reference to upper, middle and lower panels is less precise. There is even a nice new panel at the bottom of Figure S4 that is not really described in the text. Perhaps a line can be added between current lines 283 and 284.

Another minor point: for Fig. 2E, X-axis should probably say "differentially retained introns per gene"

(Remarks on code availability)

Reviewer #2

(Remarks to the Author)

(Remarks on code availability)

Reviewer #3

(Remarks to the Author)

The authors have addressed the main issues raised in my previous report. Regarding the validation of intron retention (IR) events, now provided in Figure S2: the results show the detection of both spliced and non-spliced transcripts for 5/5 of the introns tested, but while the levels of IR are substantial for pgn-53 or K03B4.4, they are rather minor for mbl-1, pct-1 or mec-8. Some levels of non-spliced transcripts are expected even for non-regulated ("fully spliced") introns because RNA isolates will also capture transcripts in the process of transcription/processing. How do we know that these events are regulated? Is the difference between the levels of IR detected by RNA-seq for these genes in different neuronal types recapitulated by RT-PCR assays?

(Remarks on code availability)

Version 2:

Reviewer comments:

Reviewer #3

(Remarks to the Author)

I am happy with the revisions and I recommend publication in Nature Communications.

(Remarks on code availability)

We thank the reviewers for their constructive suggestions. We believe we have successfully performed all the requested analyses, experiments, and modifications, as detailed below. In particular we feel that the treatment of differential intron retention and of pan-neuronal genes with cell-specific isoforms is substantially improved.

Reviewer #1

Main points:

1. Even though the authors did not generate the data, it is important that the libraries be described at the beginning, to understand what type of RNA sequencing data this was. Specifically, that this is full length, ribosome-depleted, short read sequencing, etc. This information has now been added to the Results and Methods section in the appropriate places (Lines 66-67 and 480-484).
2. One of the main points in our opinion is that, primarily for the intron retention cases, but also more generally, the authors don't really discuss what is the consequence of these events for protein production. In the case of intron retention, would this most often lead to truncated proteins? This is important information to even begin to interpret the relevance of these events. A more general analysis of how these events impact protein output is essential. We have now added more information about the coding-sequence relevance for the alternative splicing highlighted in the paper, which we hope is a significant improvement for the reader. Such examples appear in lines 82-83 (for *unc-31*), 129-132 (for intron retention), 194-195 (for *mec-2*), 199-201 (for *dgk-1*), 215-217 (for newly-added pan-neuronal gene *unc-104*), and lines 308-309 (for *unc-40*).
3. The authors do not validate any of the intron retention events in vivo. This is somewhat problematic given the high number of intron retention events the authors report, and how this is different from the number identified by Weinreb et al. The authors also comment on how their numbers are different and attribute this to the use of different algorithms. However, from both papers, it is unclear whether the authors think one approach has more false positives or false negatives. Perhaps the authors can intersect both sets of called events and collectively validate some of these events. Based on the helpful suggestions from all reviewers, we further investigated the differential intron retention we observed. First, we selected a handful of introns that were deemed alternatively spliced in our pipeline – some of which were annotated and represented in the Weinreb et al. analysis, and some of which were un-annotated and thus not present in Weinreb et al. We then performed RT-PCR of five such introns, which revealed both spliced and intron-retained bands for all five introns in question (Lines 153-157, Figure S2). This successful validation rate of both annotated and novel intron retention events lends further credence to the notion that these retained introns are biologically real. We have also added text in the Discussion section (Lines 420-427) speculating that the main reason for different intron retention results between our analysis and Weinreb et al. is that our pipeline identifies un-annotated intron retention events, and indeed, the vast majority of differential intron retention events we identify fall into the un-annotated intron retention class.
4. One of the main interesting points the authors highlight is the identification of pan-neuronal genes that have cell-specific expression patterns. However, in its current form this analysis is anecdotal, with only a couple of examples being presented. It seems like the authors could do a more global analysis with the available data: e.g. take genes expressed pan-neuronally and display their splicing uniqueness index in a

heat map; or plot the correlation between expression uniqueness and splicing uniqueness; or some other form of exploring this more systematically. We have now added such a systematic analysis of unique alternative splicing in pan-neuronal genes. First, we defined pan-neuronal genes as expressed in 123 of the possible 126 possible neuron types in the single-cell CeNGEN sequencing dataset (this was the median number of neurons in which the pan-neuronal genes from Stefanakis et al 2015 were expressed in). We then generated a “top most unique splicing pattern” heatmap only consisting of such pan-neuronal genes. Many pan-neuronal genes show highly specific splicing patterns (Fig S1, Lines 210-218 and 522-529), and a few are highlighted (see also next point).

5. Related to the specific examples that they showcase for panneuronal genes that are cell-specifically alternatively spliced, *unc-31* is indeed panneuronal (the authors could cite Stefanakis et al 2015 which is probably the most thorough examination of this expression pattern). For *dgk-1*, the evidence is less obvious from the cited study, and in fact, in the single cell cengen data this does not look panneuronal. The authors could examine this more directly or choose another better example. The validation for the cell specific *dgk-1* splicing is somewhat misleading as the authors drive the reporter with a panneuronal promoter (*rgef-1*) as opposed to using the endogenous promoter. We are grateful for the recommendation to treat the pan-neuronal observation in a more systematic way, which we think has now improved the point we are trying to make. We have made the following changes: first, we highlight *unc-31* as a pan-neuronal gene, and cite the Stefanakis et al 2015 study. We also downgrade *dgk-1* to a “broadly neuronally expressed” gene, and point out that we use a heterologous pan-neuronal promoter to drive our *dgk-1* splicing reporter. Most importantly, we generate a pan-neuronal gene uniqueness heatmap – selecting only pan-neuronal genes as defined by single-cell CeNGEN data, taking the median number of cells for which expression is detected for the gold standard pan-neuronal genes presented in Stefanakis et al as a cutoff (Lines 210-218 and 545-554). The resulting heatmap provides many potentially interesting pan-neuronal genes with cell specific isoforms (Figure S1). We then highlight one such example: *unc-104*, a pan-neuronal gene which has a cassette exon that is expressed only in two neurons.

6. The prevalence of upstream 3'ss selection in OLL neurons would also benefit from some additional analyses. Specifically, it would be important to show a distribution of % upstream usage to understand whether *rskn-1* is an outlier or whether other mRNAs are also preferentially spliced with high percentages. And second, the consequence for the proteins is not sufficiently explored. In the *rskn-1* case addition of two amino acids may or may not be consequential, and how about for other events? Do they mostly maintain frame and add a small number of aa? or do they cause early stops?

We have now performed these additional analyses. We provide a distribution of % upstream values in Fig S4, as well as an analysis of the protein-coding consequences of these alternative 3' splice sites. Unlike the intron retention results, these alternative 3' splice sites largely are frame-preserving and lack stop codons, and the majority result in a small number of alternative amino acids being encoded.

7. The authors highlight the application of the “uniqueness index” they calculate but there is one application that does not seem as developed as it could be, and would be useful if it was: instead of generating a neuron centric uniqueness index as they do for ASK in Fig. 5A, could the authors generate a gene centric, ranked table, that would also make it easy to see in which neurons a particularly specific gene is expressed in? This would be a useful resource to find neuron specific/restricted genes more easily. In addition, given that so many reporters already exist and have been analyzed with single neuron

resolution, this would serve as validation for the utility of their index. We agree that this is a nice idea. We now provide such a gene-centric analysis. The full data in table form is spreadsheet S2, and we also provide some interesting “use case” examples in heatmap form in figure S4.

Minor points:

1. Microscopy images should be zoomed in to be able to see cell bodies and projections more clearly, they are all a bit blurry. We have now provided additional zoomed in vignettes where possible to clarify the cell body positions (Fig S2, lower panel; Fig S6, upper panels; . We also clarify (lines 377-378) that in most cases, the splicing reporters contain an artificial NLS, and thus the projections should not be visible for the most part. This way the cell bodies can be specifically identified. That said, there is some degree of “leakage” and thus some processes are visible on occasion.

2. For the pqn-53 retained intron event, perhaps the authors could verify in vivo with an appropriate fluorescent reporter. We have now added in vivo verification for pqn-53, as well as a handful of other predicted retained introns, using RT-PCR (further explained above in point 3). We like the idea of further trying to validate with a reporter system, although such a system has not yet been described for intron retention. We have considered how to do so using bicolor reporters such as the ones we use here for cassettes and alternative 3' splice sites. The major challenge we faced is that the intron retained isoform invariably is frame-shifting and encodes numerous stop codons, necessitating quite a bit of re-coding, which then leaves us with the question of whether we've disrupted any important cis elements in the process. As such, for now we limited ourselves to RT-PCR validation, as well as the original evidence from single-cell and whole-animal Rna Seq.

3. In paragraph 3 of the “Upstream 3' splice site selection in OLL sensory neurons”, there is a typo in “ribosomal” Fixed!

4. Perhaps the authors could comment on the sequences preceding the upstream 3' splice sites they identify? Do they follow the consensus “UUUCAG” motif seen for *C. elegans* 3'ss?

We have now added an analysis of this question. For OLL-specific alternative 3' splice sites, we find that the downstream splice site harbors the canonical UUUCAG motif (Fig S4), while the upstream site does not. This phenomenon is not specific to OLL-specific splicing, however, as the phenomenon is also the same for all alternatively-spliced 3' splice sites (Fig S4, lower panels)

5. For the cassette exons identified with the strongest uniqueness values (unc-40 and lgc-31), are there consensus RNA-binding motifs associated with RBPs identified to regulate the splicing of these exons (SUP-12 for the unc-40 cassette exon and MEC-8 for the lgc-31 cassette exon)? If so, are these motifs present in the respective transcripts? We have now performed this analysis, and we indeed find predicted cis elements for SUP-12 upstream of the unc-40 cassette exon and MEC-8 upstream of the lgc-31 exon (Figure S6). Thus there is circumstantial evidence that this regulation is direct.

6. In the top panel of Fig. 6C, where is the AVM neuron? In general, the micrographs would benefit from outlining the worm and the cells of interest, and providing close-up views. In this particular micrograph, the AVM neuron is not in the Z plane of focus – it is difficult to visualize both the PVM and AVM in this splicing reporter simultaneously, due to their differing positions (far right lateral vs. far left lateral). This is

not a problem from ALMs or PLMs since they are bilaterally symmetrical. More broadly, we have tried to increase the demarcations showing the cell bodies of interest, and provide additional zoomed vignettes to clarify cell body positions throughout.

7. In the last sentence of the Figure 1 description, there is a typo "between gene expression and alternative splicing" Fixed!

Reviewer #3 (Remarks to the Author):

1. While the authors' approach takes advantage of the unique uniform developmental plan of *C. elegans*, it is not possible to discern, in the cases in which exons/introns/splice sites are not fully used or fully skipped, that individual cells do not have a distribution of PSI values because at the end the analysis is carried out in a population of cells (arguably identical, but that is exactly the point, they may not be so). It would be would to discuss this limitation in the text. Thank you for pointing out this important caveat, which we forgot to mention in the text. We have now added this in (Lines 341-346) where we also argue that the splicing reporters nicely complement the single cell sequencing data, answering additional questions that the seq data cannot, including the question of variability across individuals.

2. Figure 1A (and other similar analyses): the results are very interesting but no clear structure emerges from the pairwise comparisons. It may be useful to run PCA analyses to see whether there are clusters of neurons that differ more than others in differential splicing patterns (general or type-specific) and whether such clusters bear any relationship with other developmental, functional, anatomical, etc. criteria. We have now performed PCA analysis (Figure S1, text Lines 103-105), specifically comparing PCA run on gene expression differences compared to PCA run on alternative splicing differences. Both yield nice separation (reassuringly, cell type replicates cluster together), but with different patterns of separation. For example, gene expression PCAs mostly stratify sensory neurons versus all other neurons; while alternative splicing PCAs yield two distinct sensory neuron clusters, as well as a cluster of certain motor neurons (specifically, ventral cord motor neurons). These differences dovetail nicely with the observation in Figure 1 that gene expression differences are largely orthogonal to alternative splicing differences at the single-neuron level.

3. Figure 1G: the message emerging from these analyses is quite intriguing, but intron retention (the largest category of differential events according to the results of Figure 2) seems to be much less "coordinated" with other splicing types, particularly with mutually exclusive alternative exons. What do the authors make out of this? We speculate (Lines 98-101) that, like the regulation of gene expression vs. alternative splicing, the regulation of intron retention is somewhat orthogonal to the regulation of these other gene expression regulatory regimes. This also dovetails nicely with our new analysis of intron retention (Lines 129-131) which indicates that the vast majority of intron retention events result in frame-shifting and/or premature stop codon bearing transcripts, in contrast with other types of alternative splicing, and thus both the function and regulation of intron retention appear to be in a class of its own.

4. The results of Figure 2 are quite striking, but it would be good to validate the extent of intron retention in a few genes / neuron types by an independent method (e.g RT-PCR) because this is the event type that typically shows lower validation rates. I would suggest to discuss the concept of detained introns (PMID: 25561496), which correspond to single introns that are developmentally regulated, remain associated to the transcription site and are removed upon reception of a signal to allow rapid induction of gene activity. This might be the case for at least some of the intron retention events identified by the authors. In this regard, how often more than one intron retention events are observed within the same gene? Figure 2C: do the numbers in the key of the heatmap correspond to the number of intron retention events? We have now performed RT-PCR analysis on some of the top-ranked "Unique" intron retention events, and found evidence for both intron splicing and intron retention for all introns tested (5/5), which bolsters our thought that these are real biological phenomena (Figure S2). We also have added some treatment of the detained intron concept (Lines 425-429), and that this might indeed be an under-estimated phenomenon, based on our report. We added a figure panel (Figure 2) analyzing how often an intron retention event occurs in 1, 2, 3, etc. introns of a given gene, finding that the vast majority occur one intron per gene. Finally, we clarify in the panel 2C, that indeed the heat map is showing the number of intron retention events.

5. The authors convincingly argue about the value of uniqueness indexes, but it would be interesting to know more about the general distribution of uniqueness index values across neuron types and alternative splicing events. The authors are in a unique position to dive into these data. For example, what fraction of neuron types harbor highly distinct uniqueness indexes for particular classes of alternative splicing events. Is the case of OLL neurons and alternative 3' splice sites quite unique? As in point #2 above, is there a structure emerging from the data? We now address this question (Lines 260-263), and find that no other cell type has such an outlier-level of unique alternative splicing types in our data. As such, this phenomenon seems largely restricted to 3' splice sites in OLL, and for the time being we have to conclude (Lines 262-263) to be unsure about the regulatory or biological significance, or whether this phenomenon is likely to be found in other systems. Thus, although specific genes and exons have high uniqueness index values in specific cell type, there does not seem to be a trend of single cells with systematically more high uniqueness values for a particular splice type of splicing.

6. The results of Figure 6 nicely illustrate how the dataset can be used to infer mechanisms of regulation. Are other examples of alternative splicing events likely to be regulated by mec-8 or sup-12 in touch neurons? i.e. events differentially regulated in this neuron type with predicted binding sites for these factors (e.g. PMID: 26430555, PMID: 28003515). We have now performed this analysis (Figure S6) and find that indeed, there is substantial evidence for consensus binding sequences for MEC-8 and SUP-12 in many touch-neuron specific cassette exons. Moreover, we confirm that some of them have been validated as *bona fide* MEC-8-regulated cassette exons in our previous touch-neuron specific *mec-8* mutant RNA Seq data (Liang et al., 2021 NAR).

Reviewer #1 (Remarks to the Author):

The authors have done a thoughtful job addressing the points we raised. We think the manuscript is improved and this will be a very useful resource for many researchers.

A minor suggestion that we think would improve readability is to label the panels in the supplementary Figures A, B, C, etc. At the moment the reference to upper, middle and lower panels is less precise. There is even a nice new panel at the bottom of Figure S4 that is not really described in the text. Perhaps a line can be added between current lines 283 and 284.

We have now added such a sentence at line 284-285.

Another minor point: for Fig. 2E, X-axis should probably say "differentially retained introns per gene"

This has now been fixed!

Reviewer #2 (Remarks to the Author):

Reviewer #3 (Remarks to the Author):

The authors have addressed the main issues raised in my previous report. Regarding the validation of intron retention (IR) events, now provided in Figure S2: the results show the detection of both spliced and non-spliced transcripts for 5/5 of the introns tested, but while the levels of IR are substantial for pgn-53 or K03B4.4, they are rather minor for mbl-1, pct-1 or mec-8. Some levels of non-spliced transcripts are expected even for non-regulated ("fully spliced") introns because RNA isolates will also capture transcripts in the process of transcription/processing. How do we know that these events are regulated? Is the difference between the levels of IR detected by RNA-seq for these genes in different neuronal types recapitulated by RT-PCR assays?

We agree with the reviewer on the question of intron retention detectable by RT-PCR. Indeed, we are generally more inclined to trust the RNA Seq data (which is both quantitative

and normalized against the entire transcriptome) over RT-PCR (which is qualitative and normalized only internally against the gene in question). To further bolster the RT-PCR conclusions, we have made two additions. First, we clarify that, when possible, primers were designed to bind to flanking exon-exon junctions (and/or to lie on the other side of an additional very large intron) such that spliced mRNA retaining (or splicing) only the specific intron in question would yield a product (Lines 475-477). This was possible for 3/5 introns tested. Second, we now include control RT-PCRs in which introns that are not predicted to be retained to an appreciable degree are amplified under similar conditions. These control introns are not detected above baseline.

Our interpretation of the reviewer's question was: whether intron retention can erroneously be detected by RT-PCR amplification of un-processed RNA. We addressed this by (a) clarifying that the primers are designed to only amplify spliced/processed RNA, and (b) providing control experiments for introns not expected to show retention (Fig S2). We have two additional thoughts:

1. We want to clarify that the RT-PCRs proposed by the reviewer, and provided in our 1st revision, are not on individual neuron types, but on whole animals, designed as an orthogonal confirmation of the single-cell sequencing. Isolating RNA from new sorted-neuron-types for RT-PCR is not something our lab is capable of. Indeed, the cell-specific RNA Seq reported in our manuscript was generated by a consortium effort of three labs, six years, and a large NIH grant.
2. We can address the reviewer's question "Is the difference between the levels of IR detected by RNA-seq for these genes in different neuronal types recapitulated by RT-PCR assays" in one additional way: below we show a correlation of the RNA Seq average values compared to

calculated RT-PCR values (by densitometry), showing a nice correlation ($R^2 = 0.91$).

We hope that this analysis, combined with the existing requested RT-PCR analysis and control experiments, wholistically addresses the reviewer's comments.